# Nanoscale electrical conductivity imaging using a nitrogen-vacancy center in diamond

Amila Ariyaratne[1], Dolev Bluvstein[1], Bryan A. Myers[1] & Ania C. Bleszynski Jayich[1]

The electrical conductivity of a material can feature subtle, non-trivial, and spatially varying signatures with critical insight into the material's underlying physics. Here we demonstrate a conductivity imaging technique based on the atom-sized nitrogen-vacancy (NV) defect in diamond that offers local, quantitative, and non-invasive conductivity imaging with nanoscale spatial resolution. We monitor the spin relaxation rate of a single NV center in a scanning probe geometry to quantitatively image the magnetic fluctuations produced by thermal electron motion in nanopatterned metallic conductors. We achieve 40-nm scale spatial resolution of the conductivity and realize a 25-fold increase in imaging speed by implementing spin-to-charge conversion readout of a shallow NV center. NV-based conductivity imaging can probe condensed-matter systems in a new regime not accessible to existing technologies, and as a model example, we project readily achievable imaging of nanoscale phase separation in complex oxides.

[1] Department of Physics and Astronomy, University of California Santa Barbara, Santa Barbara, CA 93106, USA. Correspondence and requests for materials should be addressed to A.C.B.J. (email: ania@physics.ucsb.edu)

The motion of electrons in the solid-state provides important insight into a material's multiple interacting degrees of freedom, and understanding the complexity of these interactions is at the heart of condensed-matter physics. Measurements of a material's conductivity and its signatures as a function of temperature or magnetic field, for instance, often provide the best evidence for the microscopic mechanisms at play. In recent years, significant interest has turned to condensed-matter phenomena in which the electrical conductivity exhibits non-trivial spatial variations. Examples include topological insulators[1,2], which host conducting surfaces and an insulating interior, and Mott insulators that at critical temperatures and magnetic fields exhibit nanoscale phase separation, where pocket-like metallic regions form in an insulating matrix[3,4]. As another example, anomalous domain wall conductivity has been observed in various complex oxides, such as multiferroics[5] and iridates[6,7], with predictions of exotic Weyl semimetal behavior at these domain walls[8]. These phenomena are challenging to probe with standard transport measurements that average over a macroscopic area of the sample, and detecting these spatial signatures is the goal of many advanced techniques.

Many spatially resolved probes of local electron dynamics exist, including optical conductivity probes and scanning probe microscopy (SPM)-based probes. Optical probes are limited in their spatial resolution by the diffraction limit (with specialized near-field probes approaching resolutions of tens of nanometers[9,10]), probe conductivity at optical frequencies only, and are generally surface sensitive. SPM-style probes, such as conducting atomic force microscopy (AFM)[11] and microwave impedance microscopy (MIM)[12], offer high spatial resolution but are limited in other respects. A prominent drawback is that the signal they produce is convolved with the geometry of the probe and its interface with the sample under study, thus making quantitative measurements challenging. Importantly, all of these local conductivity probes measure the response of a system to some driving perturbation, which could induce a nonlinear response or void the subtle effects one hoped to study[13].

On the other hand, the nitrogen-vacancy (NV) center in diamond presents a quantitative, non-invasive, and nanoscale sensor capable of measuring electrical conductivity via directly sensing the magnetic fields produced by thermal electron motion in the material. An atom-like quantum sensor, the NV center's signal is analytically related to the sample conductivity through fundamental constants. And because the NV monitors the fluctuating magnetic fields produced by electrons in thermal equilibrium, the NV sensing mechanism involves neither a driving perturbation nor sample contact, which also allows for subsurface sensing. Third, due to its atomic-size, the NV center affords very high spatial resolution[14,15], with recent magnetic imaging reaching sub-10 nm resolution[16,17]. Finally, the versatility of the NV center is an attractive feature; it is capable of operating over a wide range of temperatures[18,19] and in a wide range of imaging modalities, offering the possibility of combining conductivity imaging with simultaneous magnetic, electric, and thermal imaging. Recent work by Kolkowitz et al.[20] used the NV center to sense spectral and thermal signatures of electron behavior in metallic films, elucidating transitions from diffusive to ballistic transport regimes with changes in temperature.

In this work, we integrate an NV sensor with a scanning probe microscope to spatially image local electron conductivity in nanopatterned metal films with 40-nm scale spatial resolution. By monitoring the relaxation rate of the NV center spin state while the NV is scanned in nanometer proximity to a metal, we quantitatively measure the local conductivity of several metals. Further, we demonstrate a 25-fold speedup in imaging with a shallow NV center via a spin-to-charge conversion readout

technique, which converts the easily demolished spin state into a stable charge state. With the sensitivity and spatial resolution demonstrated here, we project NV-based conductivity imaging of nanoscale phase separation in complex oxides, a model example of a spatially inhomogeneous phenomenon in a condensed-matter system.

## Results

**Relaxation model and experimental setup.** The NV center in diamond is a point defect comprising a substitutional nitrogen atom and a nearby vacancy in the carbon lattice. The two unpaired electrons of the defect center form a ground state spin triplet that features long energy relaxation times ($T_1 \sim$ ms) at room temperature. The conductivity imaging experiments described here utilize the sensitivity of the spin $T_1$ to fluctuating magnetic fields produced by electrons moving in a nearby conductor.

The Hamiltonian of the NV ground state spin triplet in the presence of a magnetic field $B$ is given by

$$H = \Delta S_z^2 + \gamma B_z S_z + \frac{1}{2}\gamma\left(B_x - iB_y\right)S_+ + \frac{1}{2}\gamma\left(B_x + iB_y\right)S_-, \quad (1)$$

where $\Delta = 2\pi \times 2.87$ GHz is the zero field splitting between the $|m_s = 0\rangle$ and $|m_s = \pm 1\rangle$ triplet states, $\gamma$ is the electron gyromagnetic ratio, $S_i$ are the spin operators and $S_\pm$ are the spin raising and lowering operators $S_x \pm iS_y$, and the $\hat{z}$ direction is chosen to point along the NV-axis[21]. Incoherently oscillating magnetic fields that are at the frequency of the $|m_s = 0 \to \pm 1\rangle$ transition and perpendicular to the NV-axis induce transitions between the two states, thus speeding up the NV center's relaxation rate[22]. This sensitivity to fluctuating magnetic fields can then be used to detect stochastic electron motion in a conductor.

The magnetic fluctuations emanating from a conductor can be related to the material's conductivity $\sigma$ by first invoking the Biot-Savart law, which gives the magnetic field produced by current densities $\mathbf{J}$ as $B_{z'}(\mathbf{r}') = \mu_0(J_x y' - J_y x')/(4\pi r'^3)$ where $\mathbf{r}' = (x', y', z')$ is the electron position and $\mu_0$ is the vacuum permeability, and then the Johnson–Nyquist formula, $S_J^{x',y',z'}(\omega) = 2k_B T \mathrm{Re}[\sigma(\omega)]$, where $S_J^{x',y',z'}$ is the spectral density of the current density fluctuations, $k_B$ is the Boltzmann constant, and $T$ is temperature. Note that $\sigma \equiv \mathrm{Re}[\sigma(\omega)] \approx \sigma(0)$ for $\omega \sim 2\pi \times 2.87$ GHz.

For the conductor geometry studied in this work, a thin film of thickness $t_{\mathrm{film}}$ a distance $d$ away from the NV sensor, a volume integral over the conductor yields the $z'$-component of the magnetic spectral density

$$S_B^{z'} = \frac{\mu_0^2 k_B T \sigma}{16\pi}\left(\frac{1}{d} - \frac{1}{d + t_{\mathrm{film}}}\right) \quad (2)$$

and $S_B^{x',y'} = S_B^{z'}/2$ (a full derivation is given in Supplementary Note 1)[23,24]. The spectral density component perpendicular to the NV-axis is, for the (100) cut diamond used in this work, $S_B^\perp = (4/3)S_B^{z'}$, which induces $|m_s = 0 \to \pm 1\rangle$ transitions at a rate $\Omega_{\mathrm{metal}}$. Applying perturbation theory to the NV center spin triplet (details in Supplementary Note 3) yields the metal-induced relaxation rate

$$\Gamma_{\mathrm{metal}} = 3\Omega_{\mathrm{metal}} = \gamma^2 \frac{\mu_0^2 k_B T \sigma}{8\pi}\left(\frac{1}{d} - \frac{1}{d + t_{\mathrm{film}}}\right). \quad (3)$$

Thus a metal induces relaxation proportional to its conductivity $\sigma$. Further, the relaxing effect goes as $1/d$ for $d \ll t_{\mathrm{film}}$ and $1/d^2$ for $d \gg t_{\mathrm{film}}$. In this work $t_{\mathrm{film}} = 85$ nm and $d$ varies from 10 to 1000 nm, thus spanning both regimes.

The conductivity imaging setup consists of a laser scanning confocal microscope integrated with a tuning fork-based atomic force microscope (AFM) (Fig. 1a)[25]. All experiments are performed in ambient conditions in a small applied magnetic field of 20 G. The AFM scans an NV center within nanometer-scale proximity of the surface of a conducting sample and the confocal microscope is used to optically initialize and readout the spin state of the NV center. Optical access is through the 150-µm thick diamond plate. A waveguide patterned on the diamond is used to transmit microwaves to coherently drive transitions between the spin states. NV centers reside ~7 nm below the surface of the bulk piece of diamond and are formed by $^{14}$N implantation and subsequent annealing (details in Methods section). To enhance photon collection efficiency, the diamond sample is patterned with 400-nm diameter nanopillars; only pillars containing 1 NV center are used here. Conducting samples are patterned onto custom-fabricated scanning probes with flat plateau-tips that have diameters of several micrometers (Fig. 1a). The fabrication procedure (details in Methods section) allows for a variety of sample geometries and materials, several of which we image in this work. The probes are then mounted onto a quartz tuning fork for AFM feedback and scanning. To minimize relative position drift between the conducting sample and the NV center, we implement temperature stabilization to ~1 mK/day in concert with active drift correction that utilizes AFM-based image registration.

**Electrical conductivity measurement.** To measure a metal's conductivity we measure the NV center relaxation rate $\Gamma_{NV} = 1/T_1$ as a function of $d$ where

$$\Gamma_{NV}(d, \sigma) = \Gamma_{metal}(d, \sigma) + \Gamma_{NV,int}, \quad (4)$$

where $\Gamma_{NV,int}$ is the intrinsic relaxation rate of the NV ($d = \infty$), which is ~200 Hz for the NVs in this study. The $T_1$ is measured by initializing the NV into its $|m_s = 0\rangle$ spin state with a 10-µs pulse of 532 nm light and then allowing the NV to decay for a dark time $\tau$ toward a thermally mixed state; this decay is measured via a subsequent spin-state-dependent photoluminescence (PL) measurement. For each $\tau$, two measurements are performed: in the first, we readout the PL of the NV $S(\tau)$; in the second, we insert a resonant microwave $\pi$ pulse after $\tau$ to swap the $|0\rangle$ and $|-1\rangle$ populations and then readout the PL $S_{swap}(\tau)$ (see Supplementary Figure 3a)[26]. The difference $S - S_{swap}$ corresponds to the difference in population between the $|0\rangle$ and $|-1\rangle$ states, which decays to 0 with $\exp(-\tau/T_1)$ as plotted in Fig. 1c (details in Supplementary Note 2). The data in Fig. 1c show a fivefold reduction in NV $T_1$ in the presence of a metal film, corresponding to $\Gamma_{metal} = 840 \pm 60$ Hz.

We now demonstrate quantitative electrical conductivity measurements using the NV center. Figure 2 plots $\Gamma_{metal}$ as a function of $d$ for three different 85-nm thick metal films of Ag, Al, and Ti, each measured with a different NV center. The films are deposited via thermal evaporation onto a 3-µm diameter flat AFM tip as depicted in Fig. 1a. Here, however, the film is continuous across the full extent of the plateau tip. Two qualitative observations can immediately be drawn from the data: first, $\Gamma_{metal}$ increases for small NV-metal separations; second, the highly conducting Ag and

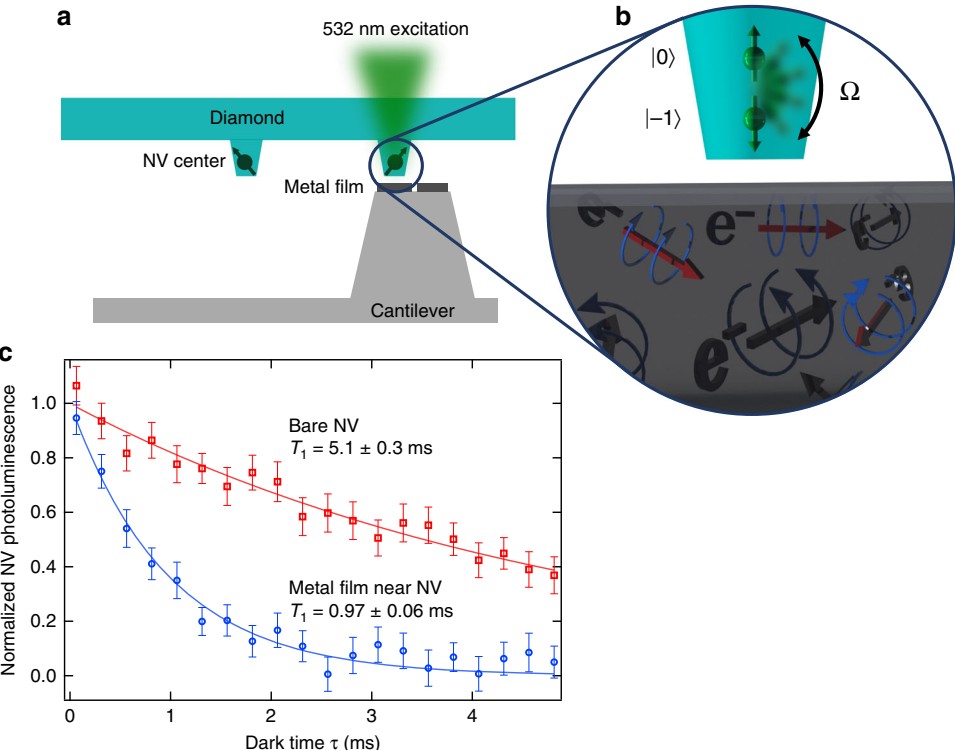

**Fig. 1** Conductivity imaging using the nitrogen-vacancy (NV) center. **a** A metallic sample is deposited onto the flat tip of an in-house-fabricated silicon scanning probe. The flat plateau region is several micrometers in diameter. This 'sample on tip' is scanned over a diamond pillar containing a single-NV center. 532 nm excitation is used for optical control and readout of the NV spin triplet. **b** Illustration demonstrating NV spin relaxation in the presence of a conductor. The stochastic, thermal motion of free electrons produces magnetic fluctuations that increase the spin transition rate $\Omega$ between $|m_s = 0\rangle$ and $|m_s = -1\rangle$, detectable as a reduction of the NV center's spin relaxation time $T_1$. **c** Measurement of the $T_1$ of an NV center far from any conductor (red squares) and positioned 100 nm above the surface of an 85-nm thick Ag film (blue circles). The specific measurement sequence is discussed in the main text and yields an exponential photoluminescence decay $\exp(-\tau/T_1)$ with $T_1 = 1/3\Omega$. The presence of the Ag film reduces the NV $T_1$ fivefold. Error bars correspond to measured standard error

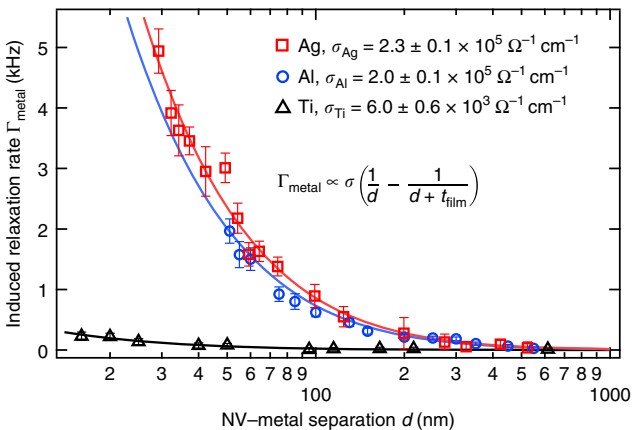

**Fig. 2** Quantitative conductivity measurements using the NV center. Plotted is the metal-induced NV center relaxation rate $\Gamma_{metal}$ as a function of NV distance $d$ from 85-nm thick films of Ag (red squares), Al (blue circles), and Ti (black triangles). For each curve the intrinsic relaxation rate of the NV is measured and subtracted to isolate $\Gamma_{metal}$. The fits (solid curves) yield the conductivity values shown. Error bars correspond to the standard error in the fit to a full $T_1$ measurement at each point

Al induce faster relaxation rates than Ti, a relatively poor conductor. These observations are consistent with Eq. 3.

We quantitatively determine $\sigma$ by performing a least squares regression on the data in Fig. 2 to Eq. 3. The extracted values of $\sigma$ are $\sigma_{Ag} = 2.3 \pm 0.1 \times 10^5 \, \Omega^{-1} \, cm^{-1}$, $\sigma_{Al} = 2.0 \pm 0.1 \times 10^5 \, \Omega^{-1} \, cm^{-1}$, and $\sigma_{Ti} = 6.0 \pm 0.6 \times 10^3 \, \Omega^{-1} \, cm^{-1}$. These values are smaller than their bulk values by factors of 2.7, 1.7, and 4, respectively. A reduced conductivity is expected for metal films whose thickness is on the order of the electron mean free path (~50 nm), and our measurements are consistent with experimental and theoretical values for 85-nm thick Ag, Al, and Ti films[27,28]. We note that Eq. 3 assumes a slab of infinite extent, but here we study slabs with ~3 μm diameter. However, this finite-size effect contributes deviations that are only ~10% of the experimental error and hence is negligible for this study (details in Supplementary Note 4).

In measuring $\Gamma_{metal}(d)$ we first contact the conducting sample to the diamond and then retract a known distance $d'$. The NV-metal separation in contact, $d_0$, is treated as a free fit parameter in the total NV-metal separation $d = d' + d_0$. Different values of $d_0$ are obtained for the three different NV-metal combinations, likely due to different tilt angles between the diamond and metal surfaces.

We also note that the data in Fig. 2 cannot be fit by a simple $1/d$ or $1/d^2$ dependence, as would be expected for $d \ll t_{film}$ or $d \gg t_{film}$, respectively, indicating that the NV's distance-dependent response is also sensitive to thickness of the conducting region.

**Nanoscale conductivity imaging**. We now demonstrate nanoscale imaging of spatially inhomogeneous conductivity by laterally scanning an NV center over an array of Al pads, as pictured in the scanning electron microscope (SEM) image in Fig. 3a. This sample is formed by thermally evaporating 85 nm of Al onto 400 nm × 400 nm pads in silicon, fabricated by etching a grid of 350-nm deep, 100-nm wide trenches in the silicon. Figure 3b plots the NV $T_1$ as the nanopatterned sample is scanned laterally over the NV. The $T_1$ is reduced when the NV is directly above the Al blocks and then recovers when above the gaps, clearly resolving the conducting features of our nanopatterned sample. The $T_1$ data in Fig. 3 are shown in Supplementary Figure 4 converted to units of conductivity. However, care must be taken to ensure that the $T_1$ features are

not an artefact of a changing $d_0$; a topographic image of the region indicates that variations in $d_0$ do not measurably affect the conductivity image (see Supplementary Figure 5).

To expedite imaging we implement an algorithm that adaptively selects $\tau$ wait times at each pixel based on the $T_1$ measurement at the previous pixel, specifically $\tau = 0$ and $\tau = 0.7 \, T_1$. We refer to this algorithm as an adaptive, single-$\tau$ measurement. This measurement method reduces the per-pixel measurement time to the order of a minute (details in Supplementary Note 6). At these time scales, however, thermal drifts can still play a significant role and we perform active NV-sample drift correction via image registration. Topographic AFM images and NV PL images can both provide highly repeatable and sharp features for drift correction. In Fig. 3b we use PL-based image registration every two hours to correct for ~10 nm NV-sample drifts with ~1 nm error (details in Supplementary Note 5)[29].

In Fig. 3c, we perform a high resolution, 5-nm point spacing line scan of the dotted orange line in Fig. 3a, demonstrating the nanoscale spatial resolution of our NV conductivity imaging technique. A different NV is used than for the image in Fig. 3b. Topographic-based image registration is performed once per hour. In addition to using an adaptive single-$\tau$ measurement, this measurement also implements a spin-to-charge readout sequence, which further reduces imaging time by a factor of 25 compared to standard spin-state dependent photoluminescence measurements and brings us to 5× the spin projection noise limit (details in Supplementary Note 7).

### Discussion

We now turn to a discussion of the spatial resolution and sensitivity of NV-based electrical conductivity imaging. From the line scan in Fig. 3c, the metal-induced magnetic fluctuations at two points separated by 5 nm is resolved within the measured $T_1$ error. However, this does not necessarily correspond to the smallest resolvable conducting feature in the material. The ultimate resolution will also depend on AFM stability and NV-metal separation. The thermally induced NV-sample drifts in Fig. 3 are ~10 nm due to infrequent drift corrections, whose frequency could be increased to minimize thermal drifts, and ultimately, picometer-scale stability could be achieved with active drift compensation techniques[30]. For the data in Fig. 3, the closest NV-metal separation $d_0$ is ~40 nm, which sets a conductivity spatial resolution of ~40 nm.

For the mechanically and thermally stable imaging apparatus used here, the NV-metal separation limits both the spatial resolution and sensitivity of conductivity imaging. The smallest achievable separation is set by the NV depth in the diamond, ~7 nm in this work. NV centers at few-nm depths that exhibit several-ms $T_1$ times have also been measured[31]. The increased NV-metal separation we observe is likely dominated by an angular misalignment between the faces of the 3-μm diameter tip and the 400-nm diameter diamond pillar; a 5° misalignment gives an NV-metal separation of ~25 nm for an NV at the center of the pillar. For the Ti data in Fig. 2, we measure $d_0 = 15 \pm 2$ nm, demonstrating the feasibility of close contact. With more controlled tilting and shallower NV centers, a 5-nm NV-metal separation and consequently a 5-nm conductivity spatial resolution should be achievable.

An important advance presented here is the implementation of NV spin-to-charge conversion (SCC) readout techniques[32] in imaging with shallow NV centers. In doing so, we significantly reduce imaging time by a factor of 25, which is particularly relevant for relaxation imaging, an inherently long measurement due to the ms-scale $T_1$ times and point-by-point scanning. Notably, we find that the SCC readout technique is highly robust

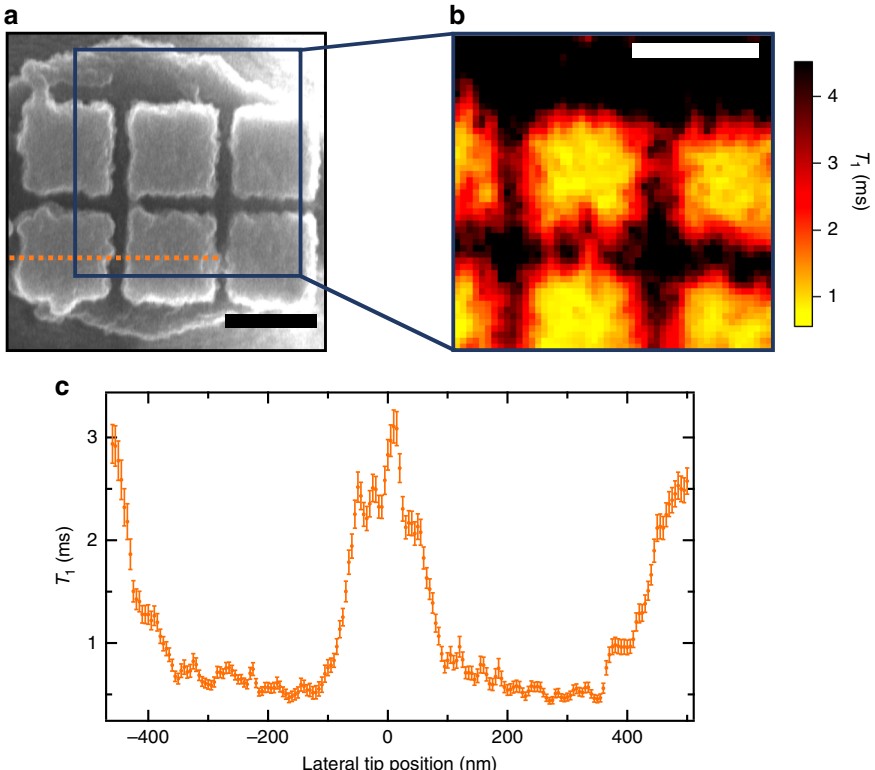

**Fig. 3** Nanoscale conductivity imaging. **a** Scanning electron microscopy image of an Al nanopattern deposited onto an AFM tip, as depicted in Fig. 1a. **b** NV $T_1$ image of the area depicted by the $1\,\mu m^2$ blue square in (**a**), produced by scanning the NV center over this area at a height of 40 nm and with 20 nm pixel spacing. **c** High-resolution $T_1$ line scan of the dotted orange line in (**a**). Features in the imaging plane are clearly resolved down to 5 nm, which is set by the point spacing in the scan. This line scan implements a spin-to-charge measurement sequence, which results in experimental measurement error only 5× the spin projection noise limit and significantly reduced imaging time: 25× faster than standard spin-dependent NV photoluminescence readout. The intrinsic $T_1$ is 6 ms for both NVs. Scale bars are 400 nm. Error-weighted, light smoothing is applied to the data in (**b**) and (**c**), for which nearest neighbors receive an additional weight reduction by a factor of 2.5. Error bars are calculated by propagating the measured standard error of the photoluminescence for the single-$\tau$ measurement of $T_1$

for shallow NV centers; all measured photostable NVs exhibit a significant enhancement in sensitivity, with a typical 20–30× reduction in measurement time (details in Supplementary Note 7). Thus SCC should find ubiquitous utility for shallow NV sensing and imaging. Implementing SCC readout and achieving $d_0 = 5$ nm yields a minimum detectable conductivity (with a signal-to-noise ratio of 1) of $10^3\,\Omega^{-1}cm^{-1} \times s^{1/2}/\sqrt{T}$, where $T$ is the total measurement time.

To illustrate the feasibility of resolving spatial conductivity variations in a relevant material system, in Fig. 4 we plot a simulated $T_1$ line scan for an NV center scanned across a material with pockets of metallic phases inside an insulating matrix, as is seen in complex oxides across a metal-to-insulator transition[33,34]. The two nanoscale metallic regions and the 5-nm wide insulating barrier separating them are clearly resolved, both in the theoretical $T_1$ curve (black line) and in the simulated measurement (orange circles). The simulated measurement accounts for the expected measurement error with 1 min of averaging per point, an adaptive-$\tau$ SCC readout technique, $d_0 = 5$ nm, an intrinsic NV $T_1$ of 6 ms, and $\sigma_{metal} = 3 \times 10^3\,\Omega^{-1}$ $cm^{-1}$. This conductivity value is typical for a Mott insulator[4], and is only half the conductivity of Ti measured in Fig. 2. To simulate the magnetic fluctuations from the conducting pockets we implement a Monte Carlo simulation of the electron trajectories (details in Supplementary Note 4)[35]. This simulation demonstrates the feasibility of non-invasive, nanoscale, NV-based imaging of inhomogeneous electrical conductivity in Mott insulators.

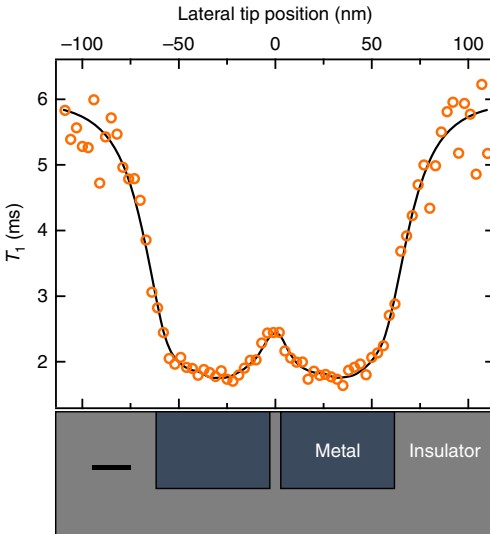

**Fig. 4** Simulated NV $T_1$ line scan taken across a Mott insulator. The Mott insulator contains conducting pockets in an insulating material, as shown schematically at the bottom of the figure. The conducting regions, with $\sigma = 3 \times 10^3\,\Omega^{-1}cm^{-1}$, are $40 \times 40 \times 60\,nm^3$ in size and are separated by 5 nm. Scale bar is 20 nm. The black curve is the theoretical $T_1$ and the orange circles represent a simulated measurement using spin-to-charge readout with 1 min of averaging per point. The NV-metal separation is 5 nm and the intrinsic NV $T_1$ is 6 ms

The NV center in diamond is emerging as a versatile quantum sensor capable of imaging magnetism[36,37], temperature[38], thermal conductivity[39], and DC currents[40,41]. In this paper, we add conductivity to the arsenal of NV imaging modalities. Future studies can, for example, combine DC magnetic field and conductivity sensing in one integrated tool to yield unique insight into materials with multiple order parameters, such as $Nd_2Ir_2O_7$ with the possibility of domain wall conductivity at magnetic domain walls[6], and buried LAO/STO interfaces with coexisting superconductivity and ferromagnetism[42].

## Methods

**Experimental setup and sample preparation**. The experimental setup consists of a home-built confocal microscope and atomic force microscope. The experiments are done in ambient conditions with active temperature control to within 1 mK. A continuous wave laser at 532 nm is used for optical pumping and readout of the NV spin, and is gated with an acousto-optic modulator (AOM). For spin-to-charge readout, continuous wave lasers at 594 nm and 637 nm gated with AOMs are also used. Photons emitted by the NV are collected into a single-mode fiber and directed to a fiber-coupled avalanche photodiode. The collection efficiency is amplified by a factor of ~5 due to waveguiding from the 400-nm wide, 500-nm tall, truncated-cone diamond pillars. Microwaves used for resonant spin manipulation are delivered via a 300-nm thick waveguide evaporated onto the diamond surface closest to the NVs. Timing of the pulse sequences is controlled by a Spincore Pulseblaster ESR-Pro 500 MHz card.

Nanopatterned metal samples are fabricated in a top-down process starting with Silicon on Insulator wafers with a 10-μm thick device layer. Nanopatterns are defined in the device Si layer using electron beam lithography and reactive-ion etching. Subsequently, a standard AFM probe fabrication process flow is carried out. The flat-faced tip is formed by stopping the KOH anisotropic Si etch such that the apex of the Si pillar retains a ~2 μm diameter with the nanopattern untouched. Cantilevers are released via a backside Si deep-etch and an HF $SiO_2$ etch. 85 nm of metal is then thermally evaporated onto the tip face.

To perform AFM, these cantilevers are then glued onto tuning forks using a micromanipulator. To minimize angular misalignments between the tip face and the diamond surface, we first use an SEM to measure the relative tilt between the tip face and the tuning fork mount. We then tilt the entire tuning fork assembly with respect to the diamond to make them parallel to within several degrees, which is limited by the SEM resolution of non-conducting silicon. Existing SEM technologies specifically designed for semiconductors could improve this resolution and tilt adjustment. We operate in tapping mode AFM, in which we electrically drive the quartz tuning fork to amplitudes of ~1 nm, and we measure the amplitude with a lock-in amplifier for feedback control (Zurich Instruments). XYZ positioning is controlled by piezoelectric scanners.

The diamond is prepared via growth of a 50-nm thick 99.99% $^{12}C$ isotopically purified thin film on a commercial Element 6 electronic grade (100) diamond substrate. Prior to growth, the diamond is etched with $ArCl_2$ plasma (1 μm) to mitigate polishing damage and cleaned in boiling acid $H_2NO_3:H_2SO_4$. NV centers are formed by $^{14}N$ ion implantation with a dosage of $5.2 \times 10^{10}$ ions/cm$^2$ at 4 keV and a 7° tilt, which yields an expected depth of 7 nm (calculated by Stopping and Range of Ions in Matter (SRIM)). The sample is then annealed in vacuum (<10$^{-6}$ Torr at max temperature) at 850 °C for 2.5 h with a 40-min temperature ramp. After annealing, the sample is cleaned in $HClO_4:H_2NO_3:H_2SO_4$ 1:1:1 for 1 h at 230–240 °C.

**Error analysis**. Errors in measured $T_1$ are given by the standard error in the exponential fit where the decay constant $T_1$ and amplitude are the only free parameters, as in Fig. 1c. In Fig. 2 a full $T_1$ measurement is done at each point and error bars correspond to the standard error in the fit. In the case of the single-$\tau$ measurement in Fig. 3, $T_1$ is explicitly calculated and the error results from propagating the measured standard error of the photoluminescence. Error in the measured conductivities is given as the standard error in the fit to the data in Fig. 2.

**Data availability**. All relevant data are available upon request from A.C.B.J.

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

## Acknowledgements

We thank Shimon Kolkowitz and Nathalie de Leon for helpful discussions. We also thank Matthew Pelliccione for constructing the scanning probe setup. This work was supported by a PECASE award from the Air Force Office of Scientific Research. A.A. acknowledges funding from the Elings Postdoctoral Prize Fellowship from the UCSB California NanoSystems Institute.

## Author contributions

A.A. fabricated the samples. A.A. and D.B. performed the experiments and data analysis. B.A.M. developed most of the experiment infrastructure. A.A., D.B., and A.C.B.J. wrote the paper. A.C.B.J. supervised the project. All authors contributed to the design of the experiment and discussions during the course of the measurements and analysis.

## Additional information

**Competing interests:** The authors declare no competing interests.

