## [Peer Review File · Nature Communications]

Reviewers' comments:

Reviewer #1 (Remarks to the Author):

The manuscript demonstrates the capability of nanoscale imaging of local conductivity using a diamond NV center. The main measurement idea is not new and has been demonstrated in Ref.20. Other experimental advances essential for the scanning probe applications such as the improved read-out methods using spin-charge conversion, the use of adaptive single-tau algorithm and drift correction have also been already demonstrated. The main novelty of this research, as I see it, is 1) putting all these advances together to demonstrate a meaningful imaging application 2) experimental demonstration of a spatial resolution with a model system 3) evaluating the applicability of the method for a problem with an interesting physics and developing the relevant modelling framework. Given the importance and the novelty of the nanoscale conductivity imaging, I believe that the manuscript is of interest to a broad community and could be published in Nature Comm.

Overall, the partitioning of the materials between the main text and the supplemental is appropriate. However, the authors should still work a bit on the readability of the paper if it is intended for a broad audience avoiding or better explaining the jargon of the specific narrow scientific field. For example, the meaning of the "spin-to-charge conversion readout" mentioned already in introduction can be understood only by looking into the referenced literature (not even described satisfactorily in the Supplemental). The same issue is with the "adaptive" algorithm. There might be more such shortcomings. Since the manuscript is in large part is an evaluation of the method, more clear/ distinct /general technical summary would be useful: what ranges of conductivity are detectable vs. spatial resolution vs. sample volume vs. acquisition time.

Reviewer #2 (Remarks to the Author):

The manuscript 'Nanoscale electrical conductivity imaging using a nitrogen-vacancy center in diamond' by A. Ariyarantne et al. demonstrates a new type of diamond NV-based scanning technique, namely imaging of nanoscale variations in a material's conductivity. The manuscript is written very well and presents innovative research with a significant potential impact. Being able to see local variations in the nanoscale conductivity of materials will benefit several communities, in particular quantum nanoelectronics and related fields. In addition, the authors elegantly implement the recently invented spin-to-charge readout method and an adaptive single-tau algorithm, which speeds up their measurements. They also present detailed derivations for the equations used in the paper. In principle, I can recommend this work for publication in Nature Communications. However, there is one important point regarding the main claim that I would like the authors to address.

The authors claim 'electrical conductivity imaging' on the basis of scanned measurements of the relaxation time T_1 , which they can relate to the material conductivity. However, as shown in figure 2, T_1 also depends in a non-trivial fashion on the separation between the NV center and the material, as well as on the material thickness. The separation dependence allows extracting the true separation as a fitting factor, which strikes me as very useful. In contrast, the only real 'images' in figure 3 are not converted into units of conductivity, but are presented in terms of T_1 . Since the entire surface seems to be covered by aluminum (is this true?), one is tempted to assume that the scans reveal topographical information, not variations in conductivity. The claim of electrical conductivity imaging would then be based entirely on the simulated 1D scan in figure 4. My questions to the authors are the following:

- Can you defend the claim of experimental conductivity imaging? Is it possible to convert the data

in fig. 3b into conductivity?

- Does your technique allow (in principle) to distinguish between topography and material conductivity by repeating separation measurements (like in figure 2) at every position or at selected positions?

- How much spatial information can the technique extract? Concretely, is it problematic if the thickness of a film varies?

In my opinion, failure to convert the data in fig. 3 into conductivity does NOT automatically devalue the paper if the authors can show a practical route toward overcoming the topography/conductivity convolution issue. The fact that the information gathered depends not only on the separation to the surface, but also on the material thickness, may complicate things initially. At the same time, however, this is also an intriguing feature that may ultimately be very useful.

Minor questions:

- Is there any danger of laser-induced heating of the sample?

- In fig. 3c, one can observe 'features' at the sampling step level of 5 nm. Later the authors explain that the smallest resolvable sample should be at roughly 40 nm. What then causes the 5 nm changes? Are these simply drifts? If understood, this may be worth explaining. If not, one has to be careful with the wording to avoid an unrealistic claim.

Reviewer #3 (Remarks to the Author):

The manuscript „Nanoscale electrical conductivity imaging using a nitrogen-vacancy center in diamond” by Amila Ariyaratne et al uses for the first time NV color centers in diamond to image the conductivity in a nanostructured metal sample. The application shown is thus very novel and very interesting the theoretical treatment of the application is scientifically sound and extensive. The method is not novel: scanning NV techniques at ambient conditions have been presented before. However, the authors combine it with a novel approach for NV spin read-out the so-called spin to charge conversion which strongly enhances the speed of the scanning imaging and must be for sure considered a significant improvement of the method.

Thus, the paper is clearly scientifically sound, except some minor points, and interesting for a specific community. However, I am not fully convinced that the research in this stage is suitable for a top-notch Journal like Nature Communication. Let me detail this statement: The authors indeed show a measurement of 400x400 nm squares of metals. However, within these squares one would not expect any nanoscale features of the conductivity and the imaging does also not resolve any just the structure of the sample with very strong edge effects. Consequently, the method cannot really prove here that it is able to resolve nanoscale conductivity features. In this context also, the discussion of the resolution remains very elusive (details see below) and I would also not trust the estimated best resolution that could be achieved (details see below). The authors discuss that the method would be highly interesting for topological insulators and complex oxides. They show a calculation on that but no results. I also raise the question if the method is versatile enough to be able to be applied to topological insulators and complex oxides: The paper does not present a real “scanning NV” technology: The scanning and the AFM feedback are reached by bringing the metal structures, and thus the sample not the probe, onto a tip like structure and onto an AFM feedback device (tuning fork). I am not sure if the processes necessary for that can be applied to complex oxides and topological insulators.

Thus summarizing, the paper presents novel results on applying NV scanning probe imaging, however, too me it is not fully clear if this really presents an advance in characterizing solid state effects, as so far it has not been applied to any “interesting samples”. I would thus at least ask the

authors to address these points more clearly and outline the potential of the method in a less “theoretical” way.

Detailed comments:

- The authors do not address the point of quenching by metals: very close to a metal, the NV quantum efficiency will significantly reduce, this restricts the minimal distance at which the NV is still usable above a metal. In the supplementary the authors claim a distance of 10-20 nm due to the loss of fluorescence. To claim this, you must measure the NV lifetime and compare to simulations just the fluorescence rate seems not appropriate here. Can you include this discussion? Does it limit the NV sensing capability e.g. close to topological insulators?
- I was wondering: is the thermal motion of the electrons the only case of T1 reduction in this technique? Are there also effects of a varying paramagnetic defect density or a varying density of electrons? For the metal this seems clear, however, for a complex oxide there might be other effect. The NV is not only sensitive to the conductivity. This might need to be included into the discussion.
- Can the authors comment on why the T1 reduction has been used and not the T2 reduction? Is it more sensitive to the effects observed here?
- Please re-phrase the description and the labeling of Fig 1c: When reading exponential fluorescence decay, I thought of the lifetime measurement of the electronic excited state of the NV. Also it was highly misleading that you call the y-axis “normalized NV PL”, this would suggest that NVs in a statistical spin mixture do not emit PL, which is not the case. You might directly label this as the population difference as described in the text.
- To judge how good the imaging technique presented here is in comparison to previous results it would be important to know: How many photons can be detected from your single NV in the pillar?
- Also unclear: you speak of the diameter of the pillar to be 400 nm. However, the pillar is not cylindrical so to which diameter do you refer, on the substrate side or on the air side?
- The discussion of the resolution is elusive, even close to scientifically unsound. What do you refer to saying that you can “resolve metal induced fluctuations at point separated by 5nm”? Do you expect any features with that size in the metal? Just that the measurement at two points can differ does not mean you are resolving something here. Also, you might want to discuss the effect that occur at the edge of the metal square.
- You mention a single tau algorithm in the text. Is this tau the waiting time for the T1 measurement? If yes, why is the tau=0 point important?
- You use a very bulky scanning probe and a large pillar. Consequently, the resolution in the topography image will be very bad. Can you detail a bit more in the text how it can nevertheless be used for drift compensation?
- I was very surprised that you use SEM to check the misalignment between probe and sample. Does this not contaminate the diamond? Is this necessary before every measurement? How can the whole setup go into the SEM while being attached to the AFM assembly? Or if not is there additional misalignment by mounting the tuning fork to the AFM nanopositioner?

Response to Reviewer #1:

The manuscript demonstrates the capability of nanoscale imaging of local conductivity using a diamond NV center. The main measurement idea is not new and has been demonstrated in Ref.20. Other experimental advances essential for the scanning probe applications such as the improved read-out methods using spin-charge conversion, the use of adaptive single-tau algorithm and drift correction have also been already demonstrated. The main novelty of this research, as I see it, is 1) putting all these advances together to demonstrate a meaningful imaging application 2) experimental demonstration of a spatial resolution with a model system 3) evaluating the applicability of the method for a problem with an interesting physics and developing the relevant modelling framework. Given the importance and the novelty of the nanoscale conductivity imaging, I believe that the manuscript is of interest to a broad community and could be published in Nature Comm.

We thank the reviewer for their endorsement of our manuscript.

Overall, the partitioning of the materials between the main text and the supplemental is appropriate. However, the authors should still work a bit on the readability of the paper if it is intended for a broad audience avoiding or better explaining the jargon of the specific narrow scientific field. For example, the meaning of the “spin-to-charge conversion readout” mentioned already in introduction can be understood only by looking into the referenced literature (not even described satisfactorily in the Supplemental). The same issue is with the “adaptive” algorithm. There might be more such shortcomings.

We thank the reviewer for this observation, and we have made the following changes to explain specific terminology.

In the last paragraph of the introduction, we have changed the spin-to-charge sentence to read:

“Further, we demonstrate a 25-fold speedup in imaging with a shallow NV center via a spin-to-charge conversion readout technique, which converts the easily demolished spin state into a stable charge state.”

In the first paragraph of supplementary section 1.4, we added a clearer description of the principle behind spin-to-charge measurement, and then refer the reader to supplementary figure 3 for the specific measurement sequence.

“The principle behind SCC readout is selectively ionizing the $m_s = 0$ spin state into the neutral charge state NV^0 while leaving the $m_s = 1$ spin state in the negative charge state (NV^-). The charge state is then readout with a yellow (594 nm) laser that selectively excites only the NV^- - charge state.”

In the second paragraph of the Nanoscale Conductivity Imaging section, we have changed the first two sentences to read

“To expedite imaging we implement an algorithm that adaptively selects tau wait times at each pixel based on the T1 measurement at the previous pixel, specifically $\tau = 0$ and $\tau = 0.7 \sim T1$. We refer to this algorithm as an adaptive, single-tau measurement.”

Since the manuscript is in large part is an evaluation of the method, more clear/ distinct /general technical summary would be useful: what ranges of conductivity are detectable vs. spatial resolution vs. sample volume vs. acquisition time.

We thank the reviewer for suggesting a clearer technical summary. We made the following changes to the text:

The end of the second paragraph of the Discussion section (the underlined portion is new):

“With more controlled tilting and shallower NV centers, a 5-nm NV-metal separation, and consequently a 5-nm conductivity spatial resolution, should be achievable.”

The end of the third paragraph of the Discussion section:

“Implementing SCC readout and achieving $d_0 = 5$ nm yields a minimum detectable conductivity (with a signal-to-noise ratio of 1) of $10^3 \Omega^{-1} \text{cm}^{-1} * \{s\}^{1/2} / \sqrt{T}$, where T is the total measurement time.”

Response to Reviewer #2:

The manuscript ‘Nanoscale electrical conductivity imaging using a nitrogen-vacancy center in diamond’ by A. Ariyaratne et al. demonstrates a new type of diamond NV-based scanning technique, namely imaging of nanoscale variations in a material’s conductivity. The manuscript is written very well and presents innovative research with a significant potential impact. Being able to see local variations in the nanoscale conductivity of materials will benefit several communities, in particular quantum nanoelectronics and related fields. In addition, the authors elegantly implement the recently invented spin-to-charge readout method and an adaptive single-tau algorithm, which speeds up their measurements. They also present detailed derivations for the equations used in the paper. In principle, I can recommend this work for publication in Nature Communications. However, there is one important point regarding the main claim that I would like the authors to address.

The authors claim ‘electrical conductivity imaging’ on the basis of scanned measurements of the relaxation time T_1 , which they can relate to the material conductivity. However, as shown in figure 2, T_1 also depends in a non-trivial fashion on the separation between the NV center and the material, as well as on the material thickness. The separation dependence allows extracting the true separation as a fitting factor, which strikes me as very useful. In contrast, the only real ‘images’ in figure 3 are not converted into units of conductivity, but are presented in terms of T_1 . Since the entire surface seems to be covered by aluminum (is this true?), one is tempted to assume that the scans reveal topographical information, not variations in conductivity.

Yes, the entire surface is covered by aluminum. The Al in the trenches, which is 300 nm lower than the pads, contributes a negligible ~3 Hz relaxation rate in comparison to both the intrinsic 170 Hz NV relaxation rate and the ~ 2000 Hz relaxation rate induced by the Al on top of the pads in Figure 3. The 3 Hz value is calculated with the finite geometry method discussed in supplementary section 1.4.

To address the concern over topographical vs. conductivity information, we have added supplementary figure 5, a topographic AFM scan of the same region imaged in Fig. 3b, to show that our images reveal predominantly variations in conductivity. The RMS height variation is 5.1 nm and does not correlate with the T_1 features in Fig. 3b.

The claim of electrical conductivity imaging would then be based entirely on the simulated 1D scan in figure 4. My questions to the authors are the following:

- Can you defend the claim of experimental conductivity imaging? Is it possible to convert the data in fig. 3b into conductivity? In my opinion, failure to convert the data in fig. 3 into conductivity does NOT automatically devalue the paper if the authors can show a practical route toward overcoming the topography/conductivity convolution issue. The fact that the information gathered depends not only on the separation to the surface, but also on the material thickness, may complicate things initially. At the same time, however, this is also an intriguing feature that may ultimately be very useful.

We thank the reviewer for the suggestion of converting our data into conductivity. A new supplementary figure 4 converts the data in figure 3 into units of conductivity, and we have added a sentence to the main text:

“The T1 data in Fig.~3 are shown in Supplementary Fig. 4 converted to units of conductivity.”

We believe that our demonstrated ability to image (Fig. 3 and SI Fig. 4) and to quantitatively measure conductivity (Fig. 2) substantiates the claim of experimental conductivity imaging.

- Does your technique allow (in principle) to distinguish between topography and material conductivity by repeating separation measurements (like in figure 2) at every position or at selected positions?

In our measurement scheme, we simultaneously acquire conductivity and topography, thus allowing us to distinguish between the two effects. The topographic image in SI Fig. 5 shows height variations with a RMS value of 5.1 nm. At a NV-metal separation (d_0) of 40 nm, these 5 nm variations would cause ~ 10% variation in the measured conductivity (an uncertainty of about $1 \cdot 10^4 \text{ ohm}^{-1} \text{ cm}^{-1}$ at maximum points of conductivity), which is the approximate magnitude of the error in supplementary figures 4a and 4b, as noted in the caption of SI Fig. 5.

As the reviewer suggests, with greater variations in topography one would repeat relaxation-versus-separation measurements such as figure 2. This could be done at every position or at sparser, selected positions if the changes in topography occur on long length scales compared to the position step size.

- How much spatial information can the technique extract? Concretely, is it problematic if the thickness of a film varies?

By repeating separation measurements as discussed above, one could also extract spatial information on the change of film thickness. As indicated by Eq. 3 of the main text, to effectively measure t_{film} the separation measurement must extend to a separation $d \sim d_0 + t_{\text{film}}$. The spatial resolution of the film thickness measurement is thus approximately $d_0 + t_{\text{film}}$, whereas the conductivity spatial resolution is approximately d_0 . This is further understood by considering supplementary equation 2. The technique can simultaneously measure changes in conductivity, topography (d_0), and film thickness all as free fit parameters, at the expense of measurement time.

Minor questions:

- Is there any danger of laser-induced heating of the sample?

We do not observe any effects of laser-induced heating (our data does not change over a range of relevant laser powers). We note that the laser is not actually on during the NV relaxation; it is only pulsed for a few microseconds between measurements. We have even seen in other results from our lab (Nature Nanotechnology 11, 700 (2016)) that for superconductors at low temperature (a

light-sensitive system!) laser-induced heating is not a problem. If laser-induced heating were a problem for some other sample in the future, various strategies could be employed to mitigate it, e.g. total internal reflection fluorescence (TIRF) microscopy.

- In fig. 3c, one can observe 'features' at the sampling step level of 5 nm. Later the authors explain that the smallest resolvable sample should be at roughly 40 nm. What then causes the 5 nm changes? Are these simply drifts? If understood, this may be worth explaining. If not, one has to be careful with the wording to avoid an unrealistic claim.

The change in NV T_1 over 5 nm steps is indeed resolved in Fig. 3c. This 5-nm resolution *in the imaging plane* is a result of the steep gradient in T_1 between the conducting and nonconducting regions. However, our ability to resolve conductivity features in the sample plane is limited by the NV-sample separation of 40 nm, which we believe is the more relevant figure of merit for spatial resolution.

If the referee is referring to the ~40 nm bumps that occur over the conducting regions in the line scan in Fig. 3c, which are approximately the same magnitude as the measured T_1 error, then we cannot confidently pinpoint the origin of these bumps, especially since they are the same magnitude as the T_1 error. As in typical convention with NV SPM experiments we claim a 40 nm resolution of the conductivity because the NV height is 40 nm and that sets the width of the NV point-spread function.

To clarify the distinction of the T_1 resolution from the conductivity resolution, we have changed wording in the captions of Fig. 3 and in the first paragraph of the Discussion section.

In the figure caption we removed the underlined portion:

“Features in the imaging plane are clearly resolved down to a spatial resolution of 5 nm, which is set by the point spacing in the scan”

In the first paragraph of the Discussion section we removed the underlined portion:

“From the line scan in Fig. 3c, the metal-induced magnetic fluctuations at two points separated by 5 nm is resolved within the measured T_1 error, thus demonstrating 5 nm spatial resolution in the imaging plane.”

Response to Reviewer #3:

The manuscript „Nanoscale electrical conductivity imaging using a nitrogen-vacancy center in diamond” by Amila Ariyaratne et al uses for the first time NV color centers in diamond to image the conductivity in a nanostructured metal sample. The application shown is thus very novel and very interesting the theoretical treatment of the application is scientifically sound and extensive. The method is not novel: scanning NV techniques at ambient conditions have been presented before. However, the authors combine it with a novel approach for NV spin read-out the so-called spin to charge conversion which strongly enhances the speed of the scanning imaging and must be for sure considered a significant improvement of the method. Thus, the paper is clearly scientifically sound, except some minor points, and interesting for a specific community. However, I am not fully convinced that the research in this stage is suitable for a top-notch Journal like Nature Communication. Let me detail this statement: The authors indeed show a measurement of 400x400 nm squares of metals. However, within these squares one would not expect any nanoscale features of the conductivity and the imaging does also not resolve any just the structure of the sample with very strong edge

effects. Consequently, the method cannot really prove here that it is able to resolve nanoscale conductivity features. In this context also, the discussion of the resolution remains very elusive (details see below) and I would also not trust the estimated best resolution that could be achieved (details see below).

We address the reviewer's concerns on resolution in direct response to their detailed comments below.

The authors discuss that the method would be highly interesting for topological insulators and complex oxides. They show a calculation on that but no results. I also raise the question if the method is versatile enough to be able to be applied to topological insulators and complex oxides: The paper does not present a real "scanning NV" technology: The scanning and the AFM feedback are reached by bringing the metal structures, and thus the sample not the probe, onto a tip like structure and onto an AFM feedback device (tuning fork). I am not sure if the processes necessary for that can be applied to complex oxides and topological insulators.

The scanning geometry in this work is that of a NV in a pillar in bulk diamond and a tip-mounted sample, a common scanning NV geometry (see, e.g. Rugar *et al*, *Nature Nanotechnology* 10, 120 (2015) and Haberle *et al*, *Nature Nanotechnology* 10, 125 (2015)). It is true that the inverse scanning geometry (with NV on tip) can be in some cases more easily integrated topological insulators and complex oxides. However, the imaging geometry we present is nevertheless compatible with a large variety of materials; any material that can be deposited on, grown on, or even bonded to silicon lends itself easily to our geometry, including many topological insulators and complex oxides (see, for example, VO₂ deposited on Si in Joushaghani *et al*. "Wavelength-size hybrid Si-VO₂ waveguide electroabsorption optical switches and photodetectors," *Opt. Express* 23, 3657-3668 (2015)). For those samples that cannot be prepared on a tip, the conductivity imaging technique we present can also be implemented in the NV-on-tip geometry.

Thus summarizing, the paper presents novel results on applying NV scanning probe imaging, however, too me it is not fully clear if this really presents an advance in characterizing solid state effects, as so far it has not been applied to any "interesting samples". I would thus at least ask the authors to address these points more clearly and outline the potential of the method in a less "theoretical" way.

The novelty in this paper is the demonstration of a novel imaging technique with greatly enhanced imaging time. The projected capabilities, as exemplified in Fig. 4, show the potential of this method to image condensed matter samples in a new regime. And, importantly, the parameters used for the simulation in Fig. 4 are those *experimentally demonstrated* in this paper, with the one exception of a closer NV-sample approach in the simulation. Upon the reviewer's suggestion, we have now included a clearer technical summary. We made the following changes to the text:

The end of the second paragraph of the Discussion section (the underlined portion is new):

"With more controlled tilting and shallower NV centers, a 5-nm NV-metal separation, and consequently a 5-nm conductivity spatial resolution, should be achievable."

The end of the third paragraph of the Discussion section:

"Implementing SCC readout and achieving $d_0 = 5$ nm yields a minimum detectable conductivity (with a signal-to-noise ratio of 1) of $10^3 \Omega^{-1} \text{ cm}^{-1} \cdot \text{s}^{1/2} / \sqrt{T}$, where T is the total measurement time."

Detailed comments:

- The authors do not address the point of quenching by metals: very close to a metal, the NV quantum efficiency will significantly reduce, this restricts the minimal distance at which the NV is still usable above a metal. In the supplementary the authors claim a distance of 10-20 nm due to the loss of fluorescence. To claim this, you must measure the NV lifetime and compare to simulations just the fluorescence rate seems not appropriate here. Can you include this discussion? Does it limit the NV sensing capability e.g. close to topological insulators?

In supplemental Fig. 1 we claim that the large variation in fluorescence indicates a distance of 10-20 nm, but we do not believe this effect to be quenching. In the previous sentence we state:

“The spatial PL modulation results from several factors that depend on the NV’s position relative to the tip: the Purcell effect, a change in the NV’s dielectric environment, and an interferometric effect between the incoming excitation laser and the tip-reflected laser [10]”

These behaviors are explained in supplemental reference 10, but we also confirmed these effects for our specific geometry using Lumerical Finite Difference Time Domain simulations in comparison with our experimental data. The PL modulation we observe is almost entirely explained by the effects above, and we do not believe to observe metal-induced quenching or substantial lifetime changes.

Furthermore, any potential future PL quenching near surfaces can be circumvented by withdrawing the tip during optical readout of the NV, which is temporally distinct from the sensing time that occurs in the dark. Further, when using the spin-to-charge measurement scheme the charge distribution is stable on second timescales, which would give even more time to withdraw the NV from the quenching substrate to then perform charge state readout.

- I was wondering: is the thermal motion of the electrons the only case of T_1 reduction in this technique? Are there also effects of a varying paramagnetic defect density or a varying density of electrons? For the metal this seems clear, however, for a complex oxide there might be other effect. The NV is not only sensitive to the conductivity. This might need to be included into the discussion.

NV relaxation can indeed be induced by different sources. In this work, the relaxation signal is dominated by conductivity, as evidenced by the excellent agreement in Fig 2 with the expected conductivities of various metals. Paramagnetic defects, for instance, could play a role but their effect would have a different NV-sample distance dependence than what we observe. Contributions from conductivity and paramagnetic defects can be distinguished in this way. Fluctuations in electron position and density do not produce magnetic fluctuations such as those produced by velocity fluctuations. Density fluctuations would only cause fluctuations in the magnitude of the spectral density. Although the thermal electron picture is qualitatively sound, the appropriate treatment uses the fluctuation dissipation theorem and one can include effects such as a nonlocal dielectric function as calculated in Ref. 24. These corrections are negligible for the present study.

- Can the authors comment on why the T_1 reduction has been used and not the T_2 reduction? Is it more sensitive to the effects observed here?

We thank the reviewer for the insightful suggestion of using T_2 . However, the T_2 times are also susceptible to surface-induced electric noise, which in turn can be modified by the presence of a conductor (see PRL 115, 087602 (2015) where T_2 is modified by a dielectric near the surface), thus complicating the interpretation. In fact, we have investigated the T_2 effect and did not see a strong, reproducible effect, an interesting subject for future exploration.

- Please re-phrase the description and the labeling of Fig 1c: When reading exponential fluorescence decay, I thought of the lifetime measurement of the electronic excited state of the NV. Also it was highly misleading that you call the y-axis “normalized NV PL”, this would suggest that NVs in a statistical spin mixture do not emit PL, which is not the case. You might directly label this as the population difference as described in the text.

In the section labelled “Electrical conductivity measurement” on pg. 2 of the manuscript, we explain the quantity being plotted in Fig 1c. We also note that this is consistent with many other groups in the NV field and their publications. Although we are measuring the population difference, as described in supplementary section 1.3 the initialization into the $m_s = 0$ state is never 100%, and thus a naive conversion to population difference would in fact be wrong, and misleading. Measuring the exact spin distribution at time of initialization is nontrivial and requires techniques such as resolving the ~nanosecond change in lifetime between the $m_s = 0$ and $m_s = +/- 1$ excited states. Although the initialization fidelity does not affect the accuracy of the T_1 measurement, as described in supplementary section 1.2, it means we cannot convert the y-axis to population difference without substantial measurements. Therefore, we believe that the “normalized NV photoluminescence” label is the most clear presentation of the data.

- To judge how good the imaging technique presented here is in comparison to previous results it would be important to know: How many photons can be detected from your single NV in the pillar?

To address this point, we have added the following two sentences in Supplementary Note 4:

“At saturation laser powers of ~ 1 mW at 532 nm, we collect ~ 500 kCounts/s from a single NV. In this work, we typically operate at ~300 uW with ~180 kCounts/s. Under 594 nm laser excitation (used for spin-to-charge readout), we typically measure ~5 kCounts/s for 3 uW incident power.”

- Also unclear: you speak of the diameter of the pillar to be 400 nm. However, the pillar is not cylindrical so to which diameter do you refer, on the substrate side or on the air side?

We refer to the diameter at the apex of the pillar.

- The discussion of the resolution is elusive, even close to scientifically unsound. What do you refer to saying that you can “resolve metal induced fluctuations at point separated by 5nm”? Do you expect any features with that size in the metal? Just that the measurement at two points can differ does not mean you are resolving something here. Also, you might want to discuss the effect that occur at the edge of the metal square.

We discuss the effect that occurs at the edge of the metal square in supplementary section 1.4

Resolution can indeed be defined in different ways, and we have tried to explicitly address our definition. We note that we do not claim 5 nm resolution of features in the metal. As is typical for NV SPM experiments, we claim that the approximate resolution of conducting features is the NV-sample separation, which is ~40 nm in this case. As described in the first paragraph of the Discussion section in the main text, in Fig. 3c we observe 5 nm spatial resolution of the metal-induced magnetic fluctuations, as the change in T_1 at two points separated by 5 nm is resolved within the measured T_1 error. This is clearly demonstrated in the region between the metal pads. Immediately we then clarify that this does not correspond to the smallest resolvable conducting features, and that the resolution also depends on NV-sample drifts and NV-sample separation. The ~10 nm drifts do not affect this resolution substantially.

To remove the confusion between the T1 resolution from the conductivity resolution, we have changed wording in the captions of Fig. 3 and in the first paragraph of the Discussion section.

In the figure caption we removed the underlined portion:

“Features in the imaging plane are clearly resolved down to a spatial resolution of 5 nm, which is set by the point spacing in the scan”

In the first paragraph of the Discussion section we removed the underlined portion:

“From the line scan in Fig. 3c, the metal-induced magnetic fluctuations at two points separated by 5 nm is resolved within the measured T₁ error, thus demonstrating 5 nm spatial resolution in the imaging plane. However, this does not necessarily correspond to the smallest resolvable conducting feature in the material.”

• You mention a single tau algorithm in the text. Is this tau the waiting time for the T1 measurement? If yes, why is the tau=0 point important?

Yes, tau is the dark waiting time in this context and throughout the paper. To clarify this, we have modified the first two sentences of the second paragraph in the Nanoscale Conductivity Imaging section to read:

“To expedite imaging we implement an algorithm that adaptively selects tau wait times at each pixel based on the T1 measurement at the previous pixel, specifically tau = 0 and tau = 0.7 T1. We refer to this algorithm as an adaptive, single-tau measurement.”

The tau = 0 point is important because, as described in supplementary section 1.2, the photoluminescence signal $S = C * \exp(-\tau / T_1)$, where C is commonly referred to as the contrast. C and T₁ are both unknowns, and thus we require at least two tau measurements to determine T₁. This choice of tau points is further explained in supplementary section 3.

• You use a very bulky scanning probe and a large pillar. Consequently, the resolution in the topography image will be very bad. Can you detail a bit more in the text how it can nevertheless be used for drift compensation?

Although the scanning probe is large at 400 nm diameter, it is slightly tilted with respect to the sample and thus allows for higher spatial resolution topographic images. An example topographic image is now included in the supplementary information as Fig. 5. The nm-scale, repeatable topographic features that appear in the AFM scan are used for image registration. The reviewer is apt in their concern as sometimes the topography does not yield suitable features, and we then use PL-based image registration as in supplementary figure 1.

• I was very surprised that you use SEM to check the misalignment between probe and sample. Does this not contaminate the diamond? Is this necessary before every measurement? How can the whole setup go into the SEM while being attached to the AFM assembly? Or if not is there additional misalignment by mounting the tuning fork to the AFM nanopositioner?

We use SEM only to check the misalignment between the probe face and the tuning fork mount. The tuning fork mount is large (1 cm), and we do not observe or expect any additional misalignment between the tuning fork mount and the AFM nanopositioner. The diamond never enters the SEM for

alignment measurements to any component. The diamond may also be tilted by several degrees relative to the AFM nanopositioner, but this can be measured confocally and accounted for *in situ*.

Further notes:

In the process of reviewing the main text and supplement, we found 4 other minor errors which we have amended:

After describing the NV Hamiltonian in the main text, we mistakenly wrote $\Delta = 2.87$ GHz, and have now changed it to $\Delta = 2\pi * 2.87$ GHz, as the Hamiltonian is in angular units.

In the first sentence of the supplement, we changed Johnson Noise to Johnson noise (improper capitalization).

In supplementary section 1.4 we mistakenly wrote Fig. 4a of the main text, and have now changed it to Fig. 4 of the main text.

We made slight aesthetic changes to supplementary figures 1 and 2, making the width of the frames consistent with the rest of the figures.

REVIEWERS' COMMENTS:

Reviewer #2 (Remarks to the Author):

The authors have addressed most points to my satisfaction. The only remaining (and crucial) issue is the separation between topography and conductivity. The authors provide now a supplementary figure 5 showing an AFM scan of the Al surface. The AFM scan was taken with the diamond pillar, which itself has a width of 400 nm, acting as scanning tip. The image is therefore highly convoluted and in my opinion absolutely unusable to identify small local variations in topography - for example, I am unable to track the outlines of the 6 pads located within the scan range. If the authors wish to demonstrate that their conductivity image does not correspond to topography features, then an AFM image with a dedicated AFM tip is needed. If this is not possible, I would propose to leave away the topography scan and to add a disclaimer somewhere stating that topography feature may show up in the conductivity image (they obviously do in the trenches between the pads anyway). As I mentioned in my first review, having topography information in the scan is not per se a problem because there are ways to distinguish between topography and conductivity (as in Figure 2).

Response to Reviewer #2:

The authors have addressed most points to my satisfaction. The only remaining (and crucial) issue is the separation between topography and conductivity. The authors provide now a supplementary figure 5 showing an AFM scan of the Al surface. The AFM scan was taken with the diamond pillar, which itself has a width of 400 nm, acting as scanning tip. The image is therefore highly convoluted and in my opinion absolutely unusable to identify small local variations in topography - for example, I am unable to track the outlines of the 6 pads located within the scan range. If the authors wish to demonstrate that their conductivity image does not correspond to topography features, then an AFM image with a dedicated AFM tip is needed. If this is not possible, I would propose to leave away the topography scan and to add a disclaimer somewhere stating that topography feature may show up in the conductivity image (they obviously do in the trenches between the pads anyway). As I mentioned in my first review, having topography information in the scan is not per se a problem because there are ways to distinguish between topography and conductivity (as in Figure 2).

We thank the reviewer for their continued attention to the issue of topography vs. conductivity. To clarify, we do not claim that Supplementary Figure 5 shows any features corresponding to the outlines of the 6 pads. We state in the figure caption:

“The topographic features produced by the convolution of the 400-nm diameter diamond pillar and the nanopatterned sample do not correlate with the T1 features in Fig. 3b.”

Conversely, without need for a dedicated AFM tip, the topography of the sample itself is sufficiently depicted by the SEM image in Fig. 3a, where the trenches are etched to be 350-nm deep.

We are not interested in topographic features of the sample, rather we are interested in the NV-sample separation d_0 and ensuring that our T1 features in Fig. 3b of the main text (and the calculated conductivity features in Supplementary Figure 4) are not simply an artefact from a changing NV-sample separation d_0 . To clarify the significance of the figure, we added the underlined portion to the figure caption:

“The topographic features produced by the convolution of the 400-nm diameter diamond pillar and the nanopatterned sample do not correlate with the T1 features in Fig.~3b, indicating that the T1 features are not produced by a changing NV-sample separation d_0 .”

We have also added the following to the caption of Figure 3:

“See Supplementary Figure 5 for a topographic image of the region.”

As explained in our first response, the Al in the trenches, which is 300 nm lower than the pads, contributes a negligible ~ 3 Hz relaxation rate in comparison to both the intrinsic 170 Hz NV relaxation rate and the ~ 2000 Hz relaxation rate induced by the Al on top of the pads in Figure 3. We can thus neglect any metal in the trenches and consider the air between the trenches as insulating material. Because d_0 is sufficiently constant (as explained in Supplementary Figure 5), this sample is thus equivalent to the simulated material in Fig. 4 of the main text, where the insulating material in Figure 3 is air.